⊘ | **Open Peer Review** | Antimicrobial Chemotherapy | Research Article

# Cell wall nanoparticles from hyphae of *Alternaria infectoria* grown with caspofungin, nikkomycin, or pyroquilon trigger different activation profiles in macrophages

Daniela Antunes,[1,2] Rita Domingues,[1,2] Mariana Cruz-Almeida,[1,2] Lisa Rodrigues,[1,2] Olga Borges,[1,2,3] Agostinho Carvalho,[4,5] Arturo Casadevall,[6] Chantal Fernandes,[1,2] Teresa Gonçalves[1,2,7]

**ABSTRACT**  *Alternaria infectoria* causes opportunistic human infections and is a source of allergens leading to respiratory allergies. In this work, we prepared cell wall nanoparticles (CWNPs) as a novel approach to study macrophage immunomodulation by fungal hyphal cell walls. *A. infectoria* was grown in the presence of caspofungin, an inhibitor of β(1,3)-glucan synthesis; nikkomycin Z, an inhibitor of chitin synthases; and pyroquilon, an inhibitor of dihydroxynaphthalene (DHN)-melanin synthesis. Distinct CWNPs were obtained from these cultures, referred to as casCWNPs, nkCWNPs, and pyrCWNPs, respectively. CWNPs are round-shaped particles with a diameter of 70–200 nm diameter particles that when added to macrophages are taken up by membrane ruffling. CWNPs with no DHN-melanin and more glucan (pyrCWNPs) caused early macrophage activation and lowest viability, with the cells exhibiting ultrastructural modifications such as higher vacuolization and formation of autophagy-like structures. CasCWNPs promoted the highest tumor necrosis factor alpha (TNF-α) and interleukin 1 beta (IL-1β) increase, also resulting in the release of partially degraded chitin, an aspect never observed in macrophage-like cells and fungi. After 6 h of interaction with CWNPs, only half were viable, except with control CWNPs. Overall, this work indicates that compounds that modify the fungal cell wall led to CWNPs with new properties that may have implications for the effects of drugs during antifungal therapy. CWNPs provide a new tool to study the interaction of hyphal fungal cell wall components with phagocytic cells and enable to show how the modification of cell wall components in *A. infectoria* can modulate the response by macrophages.

**IMPORTANCE**  *Alternaria* species are ubiquitous environmental fungi to which the human host can continuously be exposed, through the inhalation of fungal spores but also of fragments of hyphae, from desegregated mycelia. These fungi are involved in hypersensitization and severe respiratory allergies, such as asthma, and can cause opportunistic infections in immunodepressed human host leading to severe disease. The first fungal structures to interact with the host cells are the cell wall components, and their modulation leads to differential immune responses. Here, we show that fungal cells grown with cell wall inhibitors led to cell wall nanoparticles with new properties in their interaction with macrophages. With this strategy, we overcame the limitation of *in vitro* assays interacting with filamentous fungi and showed that the absence of DNH-melanin leads to higher virulence, while caspofungin leads to cells walls that trigger higher hydrolysis of chitin and higher production of cytokines.

**KEYWORDS**  *Alternaria*, fungal cell wall, nanoparticles, macrophages, immunomodulation, DHN-melanin, chitin, caspofungin, β(1,3)-glucan, nikkomycin

Address correspondence to Teresa Gonçalves, tmfog@ci.uc.pt.

The authors declare no conflict of interest.

See the funding table on p. 17.

The impact of *Alternaria* spp., dematiaceous, melanin-containing fungi, on human health worldwide has increased mostly due to the expanding number of immunocompromised and allergic patients who become susceptible to this fungus from daily exposure to fungal spores and/or fungal particles resulting from hyphal disruption (1). These saprophytes are also important phytopathogenic agents, causing great economic losses in crops, even post-harvest (1, 2). As human pathogenic fungi, *Alternaria* spp. are opportunistic agents causing phaehyphomycosis (3, 4). Clinical manifestations of *Alternaria* infections are usually cutaneous or subcutaneous lesions (5), and the information regarding resistance to antifungals used in human health to treat *Alternaria* spp. is scarce [revised in reference (1)]. Lately, there is growing interest in the study of these fungi because, together with *Aspergillus* spp., these are strong inducers of fungal sensitization and asthma (1, 6, 7).

Among the fungal structures endowed with the ability to stimulate innate immunity response, the cell wall is the first involved in contact with host cells. The fungal cell wall has several components that act as fungal pathogen-associated molecular patterns (PAMPs), which are identified by pattern recognition receptors (PRRs). The three major cell wall components, found in most medically important fungi, are β(1,3)-glucan, chitin, and mannans (8, 9). Recently, melanin has been spotlighted by both the identification of a cellular receptor that can recognize DHN-melanin and its role in the modulation of host immune cells (10).

The composition and structural organization of the fungal cell wall is in constant flux as new information emerges (11). The cell wall fibrillar core is composed of branching β(1,3)-glucans cross linked with other cell wall polysaccharides (12). The β (1,3)-glucan is recognized by the dectin-1 receptor on the surface of macrophages and other innate immune cells (13), inducing the production of proinflammatory cytokines and chemokines (14). Echinocandins are non-competitive inhibitors of β(1,3)-glucan synthase (15); of these, caspofungin was the first to be used in human medicine. Caspofungin also regulates the transcription of gene(s) coding for the β(1,3)-glucan synthase (16–18). Chitin, another major component of the fungal cell wall, is a polymer of β(1,4)-linked N-acetylglucosamine subunits joined in antiparallel chains (19). Chitin constitutes an essential part of the fungal cell wall that is required for the maintenance of its integrity, and it is a powerful immunomodulator that induces innate and/or adaptive immunity depending on the molecular size of the chitin polymer (20). Chitin is synthesized by chitin-synthases, which are grouped into seven classes (21, 22). These enzymes are important for the fitness and virulence of fungal pathogens and are the target of chitin synthase inhibitors (nikkomycins and polyoxins). Fungal exposure to nikkomycin Z may lead to an increase in the chitin content since fungi can compensate for the inhibition of one chitin synthase with the activation of other chitin synthase isoforms (23), and simultaneously, it induces different chitin microfibrils organization (24). Some fungi produce melanin that accumulates in the cell wall. *Aspergillus* spp. accumulate melanin in the conidial wall, while in *Alternaria* melanin can be found both in the conidial and in the hyphal cell wall (2, 21, 25). In fungi, there are three types of melanin, and only one is water soluble (21, 26, 27). Melanin contributes to the higher survival and competitive abilities of melanized fungi, not only in the environment but also during infection (13, 28–32). Previously, we demonstrated that *Alternaria infectoria* increases DHN-melanin synthesis in response to the antifungal treatment with caspofungin and nikkomycin Z (21). Interestingly, in *Alternaria alternata*, it was described that enzymes of the DHN-melanin synthesis pathway are shared with the pathway leading to the synthesis of mycotoxins (33).

Immune cell recognition, phagocytosis, and subsequent killing of conidia by phagocytic cells contribute to fungal clearance (34) and the induction of a proinflammatory immune response, such as the secretion of TNF-α (35, 36). This triggers local infiltration and migration of neutrophils to the site of infection to promote further antifungal clearance. Otherwise, the germination of conidia outside of the phagocytic cells or the exposure to hyphae disables phagocytosis, leading to the activation

of extracellular killing mechanisms such as neutrophil extracellular traps (37). This demonstrates that fungal cell size matters in the response of phagocytes to filamentous fungi, with different responses being observed depending on the form and size of the fungal interacting structure (conidia or hyphae) (38, 39).

In the present work, we studied how the modification of *A. infectoria* hyphal cell wall composition with drugs and a melanin inhibitor modulates the immune response to this fungus. For that we grew *A. infectoria* in the presence of caspofungin, a β (1,3)-glucan synthase inhibitor; nikkomycin Z, a chitin synthase inhibitor; and pyroquilon, a DHN-melanin synthesis inhibitor. Owing to the difficulty of studying the interaction of macrophages with *A. infectoria* in the hyphal form, as previously reported by us (40), we took a new approach preparing cell wall nanoparticles (CWNPs) from *A. infectoria* hyphae and used these to study the effect on macrophage-like cells.

## RESULTS

### Modulation of *A. infectoria* cell wall

The cell wall components of *A. infectoria* were modulated by growing the fungus with caspofungin, a β(1,3)-glucan synthesis inhibitor; nikkomycin Z, a chitin synthase inhibitor; and pyroquilon, an inhibitor of DHN-melanin synthesis. Based on our previous studies, we selected the concentrations of pyroquilon, caspofungin, and nikkomycin that had a significant impact on the content of the cell wall components (21, 23).

Depending on the fungal growth condition, macroscopic differences were observed, namely changes in the color of the mycelia, with the control condition showing a light-brown color (Fig. 1A). *A. infectoria* grown with 50 µg/mL pyroquilon presented a lighter color (in agreement with a decrease in DHN-melanin levels; Fig. 1A), while growth in the presence of nikkomycin Z at 0.5 µg/mL turned the mycelia darker, almost black (Fig. 1A). Similarly, growth in 1 µg/mL of caspofungin also led to a darker color than the untreated control but not as strong as the one observed with nikkomycin Z (Fig. 1A). As described before, these macroscopic modifications indicate the degree of melanization since darker mycelia have more DHN-melanin accumulated in the cell wall (21).

Quantification of β(1,3)-glucan and chitin cell wall content revealed that the percentage of β(1,3)-glucan decreases in the presence of nikkomycin Z or caspofungin relative to control growth conditions (Fig. 1B). In the presence of nikkomycin, the chitin content increased. Since nikkomycin Z is a chitin synthase inhibitor, this may appear to be an apparently contradictory result. However, nikkomycin Z only inhibits some chitin synthases, causing a compensating mechanism of expression of other isoenzymes (23).

Growth in pyroquilon decreased the *A. infectoria* chitin cell wall content, but the β(1,3)-glucan cell wall content was significantly higher when compared with the control fungal culture (Fig. 1B).

### Fungal cell wall nanoparticles characterization

CWNPs were prepared from mycelia of *A. infectoria* grown under control conditions (ctCWNPs), with caspofungin 1 µg/mL (casCWNPs), nikkomycin Z 0.5 µg/mL (nkCWNPs), or pyroquilon 50 µg/mL (pyrCWNPs). The mycelia were lyophilized and, subsequently, hydrated and grounded in liquid nitrogen. The method rendered particles with a small size (representation in Fig. 2A), round shaped (Fig. 2B), with diameters between 70 and 200 nm, as measured by transmission electron microscopy (TEM; Fig. 2C). The polydispersity index (PDI) was gathered by dynamic light scattering (DLS). The PDI values are between 0 and 1, in which 0.1 represents a monodispersity; values higher than 0.1 indicate polydispersity; values between 0.1 and 0.25 indicate a narrow size particle distribution. The CWNPs obtained from *A. infectoria* grown with the inhibitors had a size particle distribution that was less polydisperse and therefore narrower. The zeta potential of these particles was slightly negative, between −11 mV and −4 mV. The data regarding the zeta potential of the CWNPs was gathered by parameters as DLS, with the metric "particle diameter" as the most common descriptor of particle size. The CWNPs showed area values (by DLS) ranging approximately from 138 to 365 $nm^2$ (Fig. 2C).

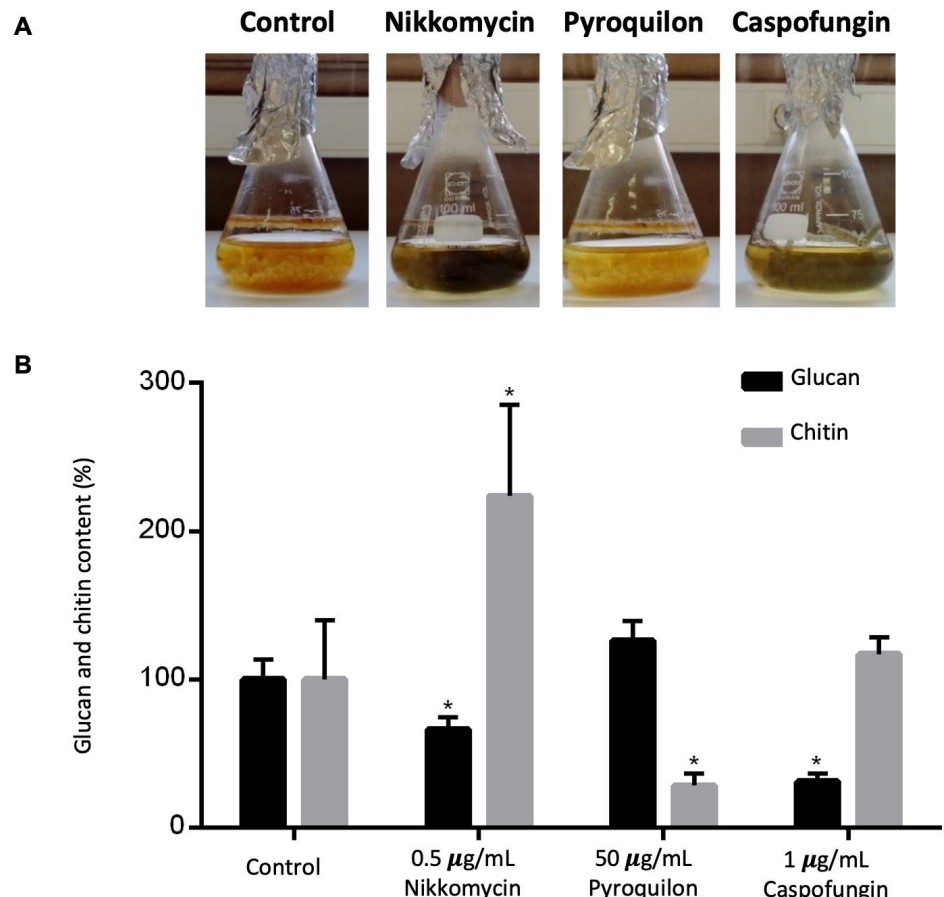

**FIG 1** Modulation of *A. infectoria* cell wall components. (A) *A. infectoria* cultured on yeast malt extract (YME) liquid media at 30°C under constant orbital shaking at 120 rpm, 3 days, with alternating 16 h light and 8 h dark periods under a blacklight lamp; growth under control conditions, or supplementation with 1 µg/mL nikkomycin, 50 µg/mL pyroquilon, or 1 µg/mL caspofungin. (B) Quantification of chitin and glucan cell wall content when grown with YME without (control) or with 0.5 µg/mL nikkomycin, 50 µg/mL pyroquilon, or 1 µg/mL caspofungin. The chitin and glucan content are normalized to the fungal biomass and expressed as a percentage of chitin or glucan content. Results are the mean ± SEM of triplicates of three independent experiments (one-way analysis of variance followed by Dunnett's *t* test post hoc analysis). *, $P < 0.05$, **, $P < 0.005$, ****, $P < 0.0001$.

Flow cytometry allows the analysis of multiple physical characteristics of single particles. The correlation of the measurements of FSC (forward-scatter light), which is proportional to the area and size of the particles, and SSC (side-scattered light), which is proportional to internal complexity, allows the differentiation of particle types and heterogeneous populations. The different growth conditions of *A. infectoria* caused differences in the CWNPs distribution parameters, acquiring different characteristics, observed by the different disposition of each CWNPs population. The results showed that ctCWNPs were more dispersed, while the nkCWNPs were less scattered (Fig. 2D).

## Interaction of CWNPs with macrophages

To characterize the interaction between macrophages and the different CWNPs, confocal microscopy was used to evaluate macrophage morphology, compartment acidification, intracellular localization of CWNPs, and quantify their uptake. For the localization of CWNPs, CWNPs were labeled with CalcoFluor White (CFW, blue) prior to the interaction assay. The quantification of CFW fluorescence after 30 min interaction did not show significant differences between CWNPs prepared from fungi grown under the different

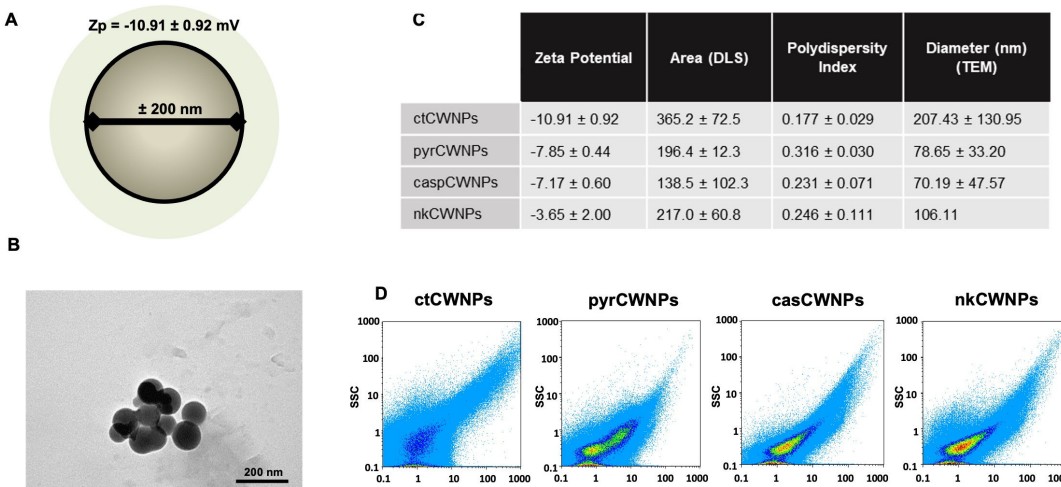

**FIG 2** CWNPs characterization. (A) Representation of a CWNP obtained from *A. infectoria* mycelia cell wall, with negative zeta potential (mean ± SD) and approximately 200 nm diameter. (B) Image of ctCWNPs obtained using TEM (scale bar = 200 nm); the images obtained were used to observe the morphology and to quantify the diameter of the CWNPs. (C) Nanoparticles' zeta potential, PDI, and area analyzed using Beckman Coulter DelsaTM Nano C Particle Analyser instrument; the diameter was quantified using TEM images. (D) Flow cytometry signatures of the CWNPs analyzed on Partec Cyflow with CWNPs labeled with fluorescein isothiocyanate (FITC). ctCWNPs, cell wall nanoparticles prepared from mycelia of *A. infectoria* grown under control conditions; casCWNPs, growth medium supplemented with caspofungin 1 µg/mL; nkCWNPs with nikkomycin Z 0.5 µg/mL; or pyrCWNPs, with pyroquilon 50 µg/mL.

conditions (results not shown), and the obtained labeling was enough to track the different CWNPs internalization and fate during the interaction assay with macrophages, with the microscopy settings remaining constant throughout the experiment (Fig. 3). In the supplemental material (see Video S1 at https://doi.org/10.6084/m9.figshare.26845231, Video S2 at https://doi.org/10.6084/m9.figshare.26849875, Video S3 at https://doi.org/10.6084/m9.figshare.26850277, and Video S4 at https://doi.org/10.6084/m9.figshare.26845252), we provide 3D projections of the macrophages with internalized CWNPs after 6 h of interaction.

To track the acidic compartments, the selective probe Lysotracker Red (red) was used. The interaction of RAW264.7 macrophages with control cell wall nanoparticles (ctCWNPs) did not result in morphological changes during the 6 h interaction (Fig. 3; see Video S1 at https://doi.org/10.6084/m9.figshare.26845231). However, macrophages exposed to pyrCWNPs during 6 h adopted a stretched morphology; the same was observed during the interaction with nkCWNPs. Macrophages incubated with casCWNPs triggered earlier morphological alterations upon 3 h of interaction (Fig. 3).

A qualitative analysis indicated that Lysotracker Red probing showed differences in acidification due to macrophage interaction with casCWNPs, nkCWNPs, and pyrCWNPs when compared with control (ctCWNPs; see Fig. S1 at https://doi.org/10.6084/m9.figshare.26850835). After 30 min of exposure CWNPs, only casCWNPs increased the acidification of the macrophage intracellular compartments; this effect was visible until the first 3 h of exposure with pyrCWNPs and casCWNPs, expressed by a brighter red fluorescence. In the case of nkCWNPs, there was no increase in red fluorescence (Fig. 3; see Video S2 at https://doi.org/10.6084/m9.figshare.26849875). Macrophages internalizing pyrCWNPs exhibited the brightest red spots (Fig. 3; see Video S3 at https://doi.org/10.6084/m9.figshare.26850277); the acidification was accompanied by a modification of the macrophage morphology (Fig. 3; see Video S3).

The confocal fluorescence microscopy also showed that casCWNPs led to the development of a blue halo (CFW) around the macrophages after 3 h interaction (Fig. 3, white arrow; see Video S4 at https://doi.org/10.6084/m9.figshare.26845252). This effect was not observed in any other condition (Fig. 3).

The quantitative analysis of the fluorescence microscopy images allowed the study of several parameters to obtain a detailed characterization of how macrophages interacted

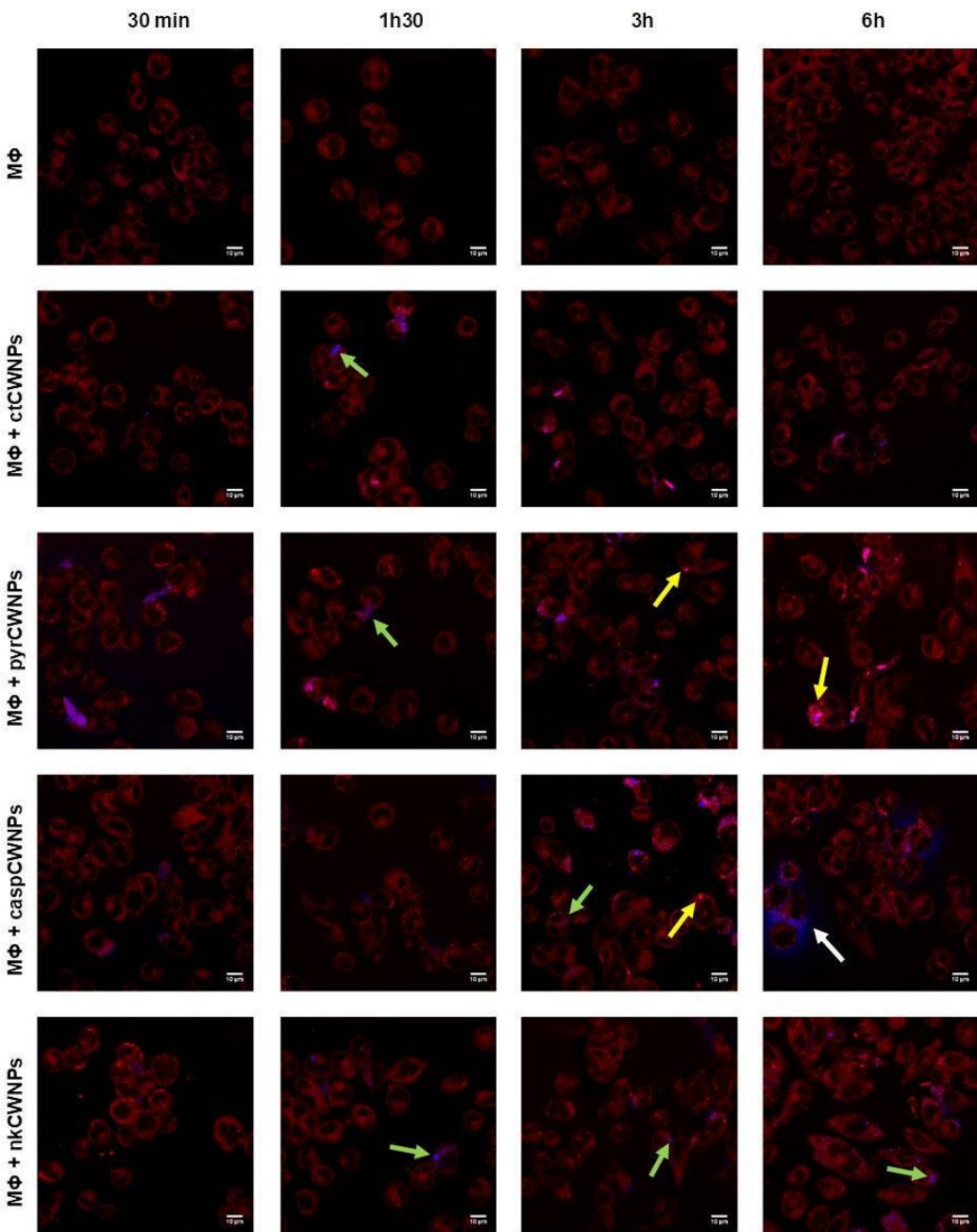

**FIG 3** Activation of RAW264.7 macrophage-like cells by CWNPs. Macrophages (MΦ) were exposed during 0.5, 1.5, 3, and 6 h to CWNPs of fungi grown in control conditions (ctCWNPs); the presence of 50 µg/mL pyroquilon (pyrCWNPs); or 1 µg/mL caspofungin (casCWNPs); or 0.5 µg/mL nikkomycin Z (nkCWNPs). The CWNPs were labeled with CFW (Fluorescent Brightener 28, Sigma), and macrophages were labeled with LysoTracker Red (Red DND-99, Invitrogen). White arrows represent a blue halo around macrophages internalizing casCWNPs. Yellow arrows indicate brighter acidified red points; green arrows on blue points (particles) do not overlap with red fluorescence. Performed on Carl Zeiss LSM 710 Confocal Microscope, using a 63× PlanApoChromat (NA 1.4) oil objective (scale bar = 10 µm).

with the different CWNPs used in this study. The internalization process of nkCWNPs and casCWNPs was more efficient than that of ctCWNPs and pyrCWNPs since the number of macrophages internalizing nkCWNPs and casCWNPs was higher, especially after 3 and 6 h of interaction assay (Fig. 4, top) when compared with ctCWNPs. Also, the number of CWNPs internalized per macrophage cell was higher with nkCWNPs and casCWNPs when compared with the control (ctCWNPs) and pyrCWNPs (Fig. 4, bottom).

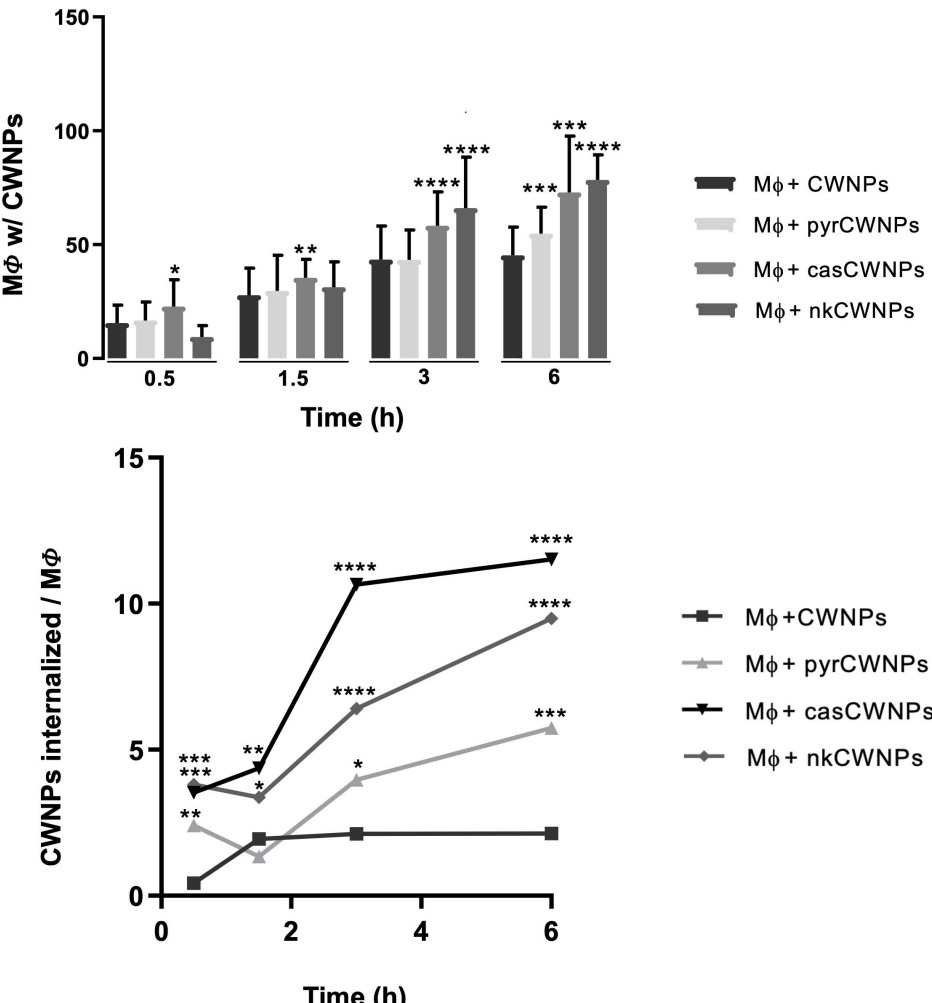

**FIG 4** Quantification of the internalization of CWNPs by RAW 264.7 macrophages. (Top) Number of macrophages (MF) with CWNPs and (bottom) average number of CWNPs per macrophages after 0.5, 1.5, 3, and 6 h. Macrophages were exposed to control cell wall nanoparticles (ctCWNPs), to cell wall nanoparticles of fungi grown with pyroquilon (pyrCWNPs), or with caspofungin (casCWNPs) or with nikkomycin (nkCWNPs). The CWNPs were labeled with CFW (Fluorescent Brightener 28, Sigma), and macrophages were labeled with LysoTracker Red (Red DND-99, Invitrogen). Confocal fluorescence microscopy was performed on a Carl Zeiss LSM 710 Confocal Microscope, using a 63× PlanApoChromat (NA 1.4) oil objective. The images were analyzed with Fiji using ImajeJ Plugging Cell Counter using at least five images (microscope field) for each condition at each time point of the interaction; counts were normalized with a total number of macrophages in each image. The threshold of 50 macrophages was used, meaning that at least 50 macrophages were considered in each count. Results are the mean ± the SEM of triplicates of three independent experiments (one-way analysis of variance followed by Dunnett's $t$ test post hoc analysis). *, $P < 0.05$, **, $P < 0.005$, and ****, $P < 0.0001$.

To further investigate how CWNPs were internalized by macrophages, TEM was performed (Fig. 5). Macrophages exhibited filopodia extensions to hold and internalize CWNPs; internalization by membrane ruffling could be observed (Fig. 5B; black arrows). Intracellular compartments delimited by a membrane and enclosing CWNPs were noticed, although some nanoparticles were free in the cytoplasm (Fig. 5B, CWNP). When compared with the macrophages alone (Fig. 5A), macrophages internalizing *A. infectoria* CWNPs exhibited more vacuoles and the rough endoplasmic reticulum became more visible (Fig. 5). Since the inhibition of the synthesis of fungal DHN-melanin by pyroquilon (pyrCWNPs) induced major changes in the macrophage morphology, appearing more elongated and with more acidic intracellular compartments (Fig. 3), we were prompted to study by TEM the major ultrastructure modifications observed in macrophages

following 3 h of interaction with the pyrCWNPs. We observed that there was an overall degradation of the macrophages, with cell burst (Fig. 5C, middle panel), high degree of vacuolization (v), and autophagy-like structures (a), with double membrane compartments (Fig. 5C, a).

## Macrophage viability

In the early interaction of macrophages with fungal CWNPs (30 min), the macrophage viability did not change for any of the conditions tested (Fig. 6A). However, longer time intervals of exposure led to a different viability pattern of the phagocytic cells, depending on the nature of the CWNPs to which they were exposed. After a 1.5 h period, only the CWNPs obtained from fungi grown with pyroquilon (pyrCWNPs), under DHN-melanin synthesis inhibition conditions, resulted in 10% decrease of the macrophages viability (Fig. 6A). After a 3 h period, all treated CWNPs showed decreased viability of macrophages, and 6 h after the beginning of the exposure assay, only half of the macrophages were viable (Fig. 6A).

## casCWNPs induce pro-inflammatory cytokines

As an indicator of the activation of macrophages, we quantified the expression of the gene coding for the TNF-α and the concentration of the cytokine TNF-α and IL-1β accumulated at the end of the assay, after 6 h of macrophages interaction with CWNPs (Fig. 6B through D). Quantitative real time PCR showed that in a population of macrophages interacting with ctCWNPs, the expression of the TNF-α gene did not change significantly (Fig. 6B). When macrophages were exposed during 3 h to CWNPs prepared from *A. infectoria* grown with several inhibitors of cell wall components, only the casCWNPs lead to a significant increase in the relative expression of TNF-α. This increased gene expression resulted in an effective release and accumulation of TNF-α in the supernatant after 6 h of exposure to casCWNPs (Fig. 6C). Nevertheless, macrophages exposed to pyrCWNPs or nkCWNPS also showed a tendency to increase in TNF-α release (Fig. 6C). The release of the IL-1β followed the same pattern of that observed for TNF-α (Fig. 6D).

Overall, our results (Fig. 7) showed that the exposure of RAW264.7 macrophages to cell wall nanoparticles obtained from fungal mycelia grown in the presence of the inhibitors leads to the following responses: (i) pyroquilon, a DHN-melanin synthesis inhibitor, prompts macrophages to readily undergo cell death with signs of autophagy but does not trigger immediate production of immune determinants such as TNF-α; (ii) by inhibiting β(1,3)-glucan synthesis with caspofungin, a prompt release of TNF-α and IL-1β was observed but no immediate cell death; a blue halo surrounding the macrophages indicates the hydrolysis of chitin inside cytoplasmic compartments; (iii) CWNPs with more chitin, more melanin, and less glucan (obtained from mycelia grown in the presence of nikkomycin) had a delayed macrophage response and internalization.

## DISCUSSION

In fungi, the cell wall is the first point of contact and thus plays a critical important role in the recognition and phagocytosis by the host cells. The fungal cell wall is a dynamic structure, and as we described previously, the use of antifungals directed against the cell wall, such as caspofungin or nikkomycin Z, leads to the modulation of the components of its structure, namely in *A. infectoria* (23). In this fungus, we also described DHN-melanin synthesis as a potential target for antifungals (21). Now we aimed to unravel the effect of the hyphal cell wall modulation of β−1,3-glucan, chitin, and DHN-melanin caused by these antifungals in the *A. infectoria* recognition by immune host cells.

Most of the studies regarding filamentous fungi cell wall recognition by host cells have been done with *Aspergillus fumigatus*. This is not surprising given the clinical importance of aspergillosis. Regarding *A. fumigatus*, macrophages ingest and kill resting conidia, while neutrophils use oxygen-dependent mechanisms to attack hyphae germinated from conidia that escape macrophage surveillance (42), and dendritic cells

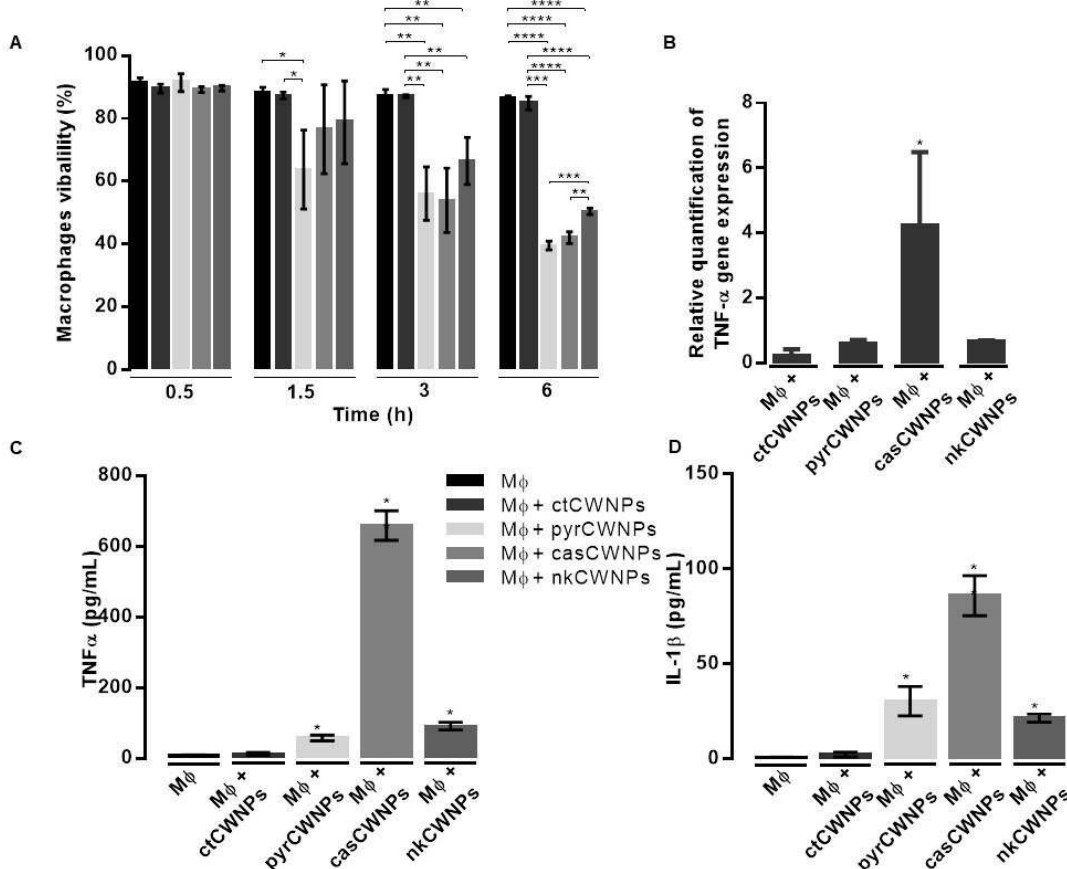

**FIG 6** Macrophage response to CWNPs stimulation. (A) Analysis of viable macrophages using the Trypan Blue staining test. Macrophages were exposed during 0.5, 1.5, 3, and 6 h to ctCWNPs and to pyrCWNPs, casCWNPs, or pyrCWNPs. Results are the mean ± SE of triplicates of three independent experiments (one-way analysis of variance followed by Dunnett's $t$ test post hoc analysis). *, $P < 0.05$, **, $P < 0.005$, and ****, $P < 0.0001$. (B) Relative quantification of the expression of the gene coding for TNF-α by the macrophages after 3 h stimulation with ctCWNPs, pyrCWNPs, casCWNPs, or nkCWNPs. The data were analyzed by relative quantification using the $2^{-\Delta\Delta Ct}$ method (41). The results represent the mean ± SEM ($n = 3$). (C) TNF-α and (D) IL-1β secretion by macrophages after 6 h interaction with ctCWNPs, pyrCWNPs, caspCWNPs, or nkCWNPs. The quantification of the cytokines was accomplished using enzyme-linked immunosorbent assay (ELISA) ($n = 2$).

also perform an important protective role (43). With regard to filamentous fungi, there are dramatic differences between the cell wall composition of conidia and hyphae (44), and consequently, the innate immune system responds differently to conidia and hyphae (34, 45). However, the hyphal form is difficult to study *in vitro* with regard to its ability to interact with cells of the innate immune system due to their size and to the fact that in contrast to conidia, these are readily recognized by the host cells (45). Some elaborate strategies were developed to overcome these limitations, including the development of microfluidic devices (46).

Previous studies from our research group have focused on the cell wall structure and its modulation by antifungals of common environmental fungi, *Alternaria* sp., with particular interest in *A. infectoria*, which is an opportunistic agent of human fungal infection and of fungal sensitization and asthma (18, 21, 23). The absence of studies regarding how *Alternaria* cells interact with macrophages led us to study the interaction of macrophages with *A. infectoria* conidia (40) leading us to the conclusion that its large conidia (7–10 × 23–34 µm) are readily internalized by macrophages, but the subsequent macrophage responses are insufficient to effectively kill conidia. However, when we initiated the studies to unravel the interaction of macrophages with *A. infectoria* hyphae, we were confronted with a major technical limitation: within a few minutes of co-incubation, the entire population of macrophages died. Also, germinating conidia during

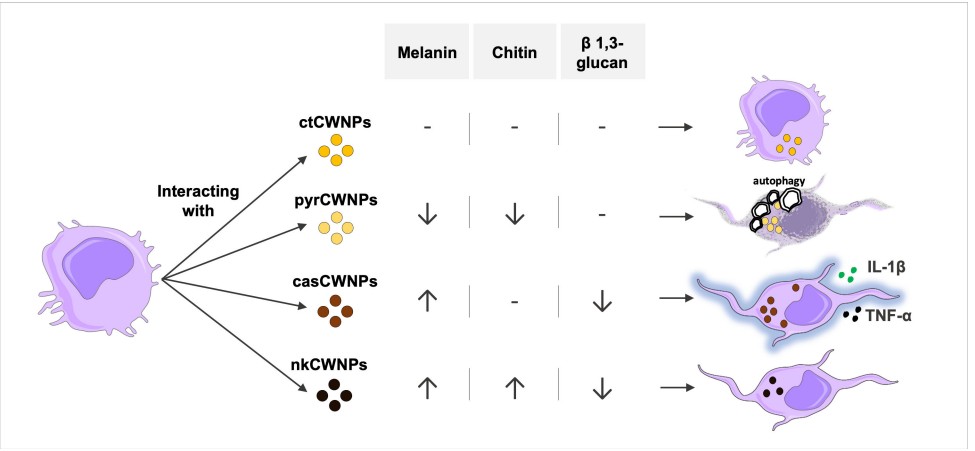

**FIG 7** Modification of the hyphal cell wall by antifungals or DHN-melanin synthesis inhibitors leads to a different early response by cells of the innate immune system.

18 h resulted in massive macrophage death (40). This prompted us to develop a new model to study how macrophages respond to *A. infectoria* hyphal cell wall. For this, we prepared CWNPs from *A. infectoria* mycelia grown with different compounds that modulate the cell wall content in chitin, β(1,3)-glucan, and DHN-melanin. The nanoparticles obtained had sizes varying from 70 to 200 nm, with negative zeta potential, and showed a tendency to aggregate, as demonstrated by TEM. Some authors contend that particles smaller than 0.5 µm are not internalized by phagocytosis (47). However, professional phagocytic cells can engulf nanoparticles by different mechanisms, including clathrin-dependent endocytosis, macropinocytosis, and phagocytosis (48). In this study, we exposed macrophage-like cells, the cell line RAW264.7, to fungal CWNPs and, using TEM, observed that the nanoparticles were taken up by the macrophages by membrane ruffling. These CWNPs could be inside compartments delimited by membranes or free in the cytoplasm. Moreover, the interaction of the CWNPs with macrophages leads to differential host cell activation depending on the CWNP type and, consequently, on the composition of the fungal cell wall. In contrast to the destructive effects of whole hyphae, the viability of macrophages upon interaction with ctCWNPs was not affected, suggesting that the hyphal effect on macrophage cell death is more likely to be a function of the size of the interacting particles (or by enzymes secreted by hyphae) and less on the cell wall composition.

*A. infectoria* grown in the presence of caspofungin leads to increased melanin content and a decrease in β (1,3)-glucan, while the chitin cell wall levels remain unchanged (21, 23), and the mycelia become darker. Paradoxically, when grown in the presence of nikkomycin Z, a chitin synthase inhibitor, the amount of chitin in the cell wall increased. We attributed this effect to the fact that *A. infectoria* is endowed with eight different chitin synthases (23), and nikkomycin Z only inhibits some of the enzyme isoforms (49); the redundancy and the compensatory effect of another enzyme(s) account(s) for the increased chitin content. The observation that when grown with nikkomycin Z the mycelia became more melanized is likely to be related to the dependency of melanin disposition and chitin synthesis (24) since chitin is required as a "scaffold" to cross-link melanin to the cell wall components (21, 24, 50, 51).

In this work, we quantified macrophage viability using a trypan blue exclusion assay. Although there are other quantification methodologies (52), and the association of flow cytometry (53) increases the accuracy of using trypan blue assay, the robustness and consistency of the results justify its validation. The early (first 6 h) interaction of *A. infectoria* conidia with RAW264.7 macrophages does not change macrophage's viability, as previously indicated by us (40). Now, it is reported that despite the ctCWNPs not influencing macrophage viability initially, after 6 h interaction, CWNPs from fungi grown

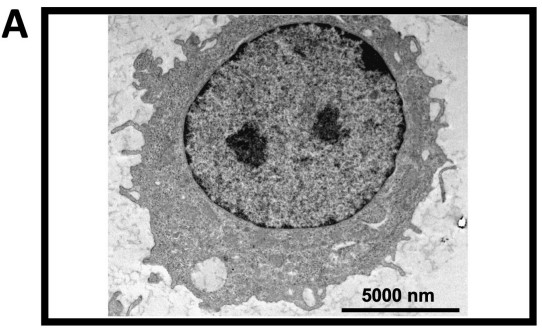

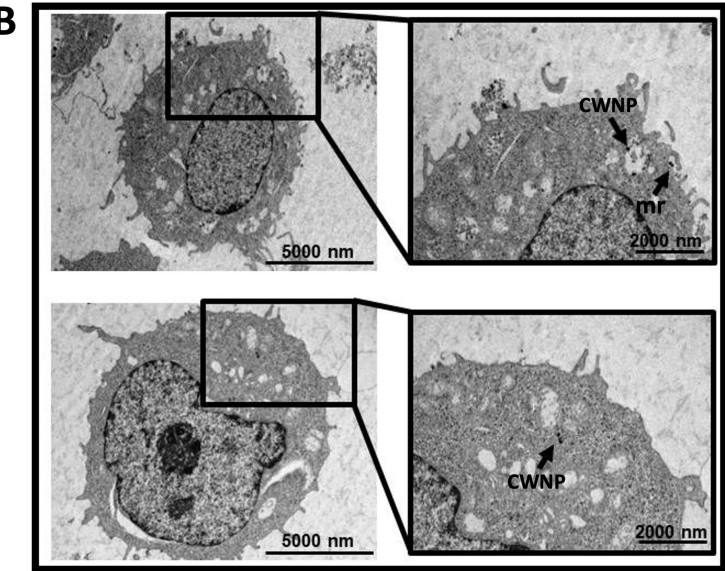

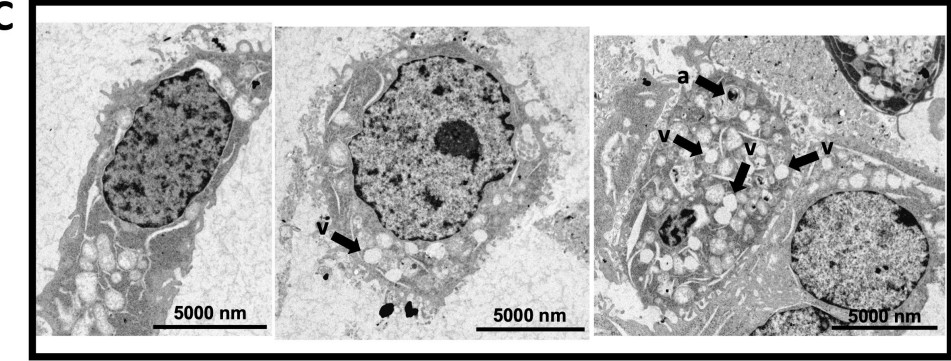

**FIG 5** TEM study of the intracellular distribution of CWNPs in RAW 264.7 macrophages and resulting cellular ultrastructure modifications. Macrophages were exposed to CWNPs during 3 h; the cells were fixed with 2.5% glutaraldehyde in 0.1 M sodium cacodylate buffer (pH 7.2). Post fixation was performed with 1% osmium tetroxide for 1 h; these cells were removed by scratching and washing. In contrast, it was used aqueous uranyl acetate (1%). After washing, the samples were dehydrated in a graded ethanol series (30%–100%), impregnated, and embedded in Epoxy resin (Fluka Analytical, Buchs, Switzerland). Ultrathin sections (~70 nm) were mounted on copper grids and stained with lead citrate 0.2% for 7 min. (A) RAW 264.7 macrophage not exposed to CWNPs. (B) ctCWNPs internalization by RAW 264.7; CWNPs (CWNP) are internalized by membrane ruffling and may persist inside intracellular compartments or be free in the macrophage cytoplasm. (C) pyrCWNPs internalization by macrophages leads to morphological alteration of the macrophages, to a high degree of vacuolization (V), and autophagy-like signs (a). The images were obtained with FEI-Tecnai G2 Spirit Bio Twin TEM at 100 kV.

in the presence of pyroquilon, nikkomycin Z, and caspofungin all reduced macrophage viability. So, the structure/components of the hyphal ctCWNPs, when interacting with macrophages, most probably behave similarly to conidia, with a delayed mild response

from the host cells. Otherwise, when the fungal hyphal cell wall structure is affected by antifungals directed to the synthesis of *A. infectoria* cell wall components such as chitin, beta glucan, and DHN-melanin, PAMPs in the resulting CWNP exert a more exacerbated response by macrophages which lead to loss of cell viability.

In addition, macrophage morphological modifications indicated a stronger activation state for all CWNPs except ctCWNPs. Although caspofungin alters the β(1,3)-glucan content by inhibiting its synthesis, the remaining chitin becomes more exposed (54, 55). Since β(1,3)-glucan is known to have a strong immunostimulatory effect (56), this could explain the observed enhancement in macrophage activation, with an increased acidified compartment required to kill the pathogen. Nevertheless, only macrophages interacting with casCWNPs showed a significant increase in TNF-α gene expression and higher production of TNF-α and IL-1β, standing for a more pro-inflammatory profile of the macrophages in this condition. TNF-α and Il-1β are central signaling molecules produced during pro-inflammatory responses and can be induced by the activation of dectin-1 by β(1,3)-glucan (57). Based on previous studies indicating the differential exposure of cell wall components depending on the antifungal used, one can expect that on the casCWNPs particles, β(1,3)-glucan is presumably more exposed, while on nkCWNPs and pyrCWNPs may not be so exposed and/or masked by melanin or chitin. Also associated with the response of macrophages to casCWNPs, we observed a phenomenon hitherto undescribed: the occurrence of a blue halo (blue fluorescence due to the label of chitin with CFW) around macrophages after 3 h of co-incubation with casCWNPs. This result suggests that the long polymers of chitin of internalized casCWNPs were degraded into smaller lengths and/or building blocks that were released to the extracellular milieu and warrants further investigation in future studies. Hydrolyzed chitin in the extracellular milieu may function as a messenger to signal infection between immune cells (58). Of utmost importance, the activity of host chitinases by generating free chitin contributes to immunomodulation, inflammation, and asthma (59). The role of chitin in the immune system has been thoroughly revised (60). Therefore, we believe this finding can be connected to macrophages producing and secreting degrading enzymes to intracellular compartments containing foreign materials (59), including glycosidases that can hydrolyze chitin. Nevertheless, the reason why this is only observed in casCWNPs is currently unexplained and deserves further studies.

For *A. infectoria,* nikkomycin Z leads to higher cell wall chitin content and, concomitantly, to higher DHN-melanin cell wall content (21, 61). We observed that nkCWNPs caused a delayed macrophage response and a delay of their internalization, which can be explained since chitin tends to be immunosuppressive, blocking dectin-1-mediated engagement (62), and DHN-melanin can be immunomodulatory. DHN-melanin masks certain components of the fungal cell wall (13, 29). As described in *A. fumigatus,* DHN-melanin is recognized by its receptor MelLec (10), and its activation contributes to the remodeling of intracellular calcium, inducing glycolysis activation through HIF-1α and mammalian target of rapamycin (mTOR) (32). The pyrCWNPs, prepared from mycelia grown with pyroquilon—an inhibitor of DHN-melanin synthesis and an inducer of lower cell wall chitin content (21)—led to higher macrophage morphological activation, to earlier macrophage death, and to the highest increase of acidified intracellular compartments. TEM also showed that the phagocytic cells ultrastructure was dramatically changed due to 3 h exposure to pyrCWNPs, showing cellular integrity compromised, double membrane autophagy-like compartments, and highly vacuolized cytoplasm. The inhibition of melanin synthesis exposes cell wall components that trigger macrophage stimulation (21, 29, 63, 64). So, the results obtained using these CWNPs are in accordance with previous studies showing that melanin plays an important role in the recognition of fungi, mainly by being responsible for the inhibition of the phagolysosome maturation (13).

Although we recognize the limitation of our surrogate, due to the small size of the nanoparticles in relation to fungal hyphae and due to the exposure of internal structures, the results obtained in the current study using CWNPs with modulation in the cell wall

components are in good agreement with prior literature observations. We recognize that it is difficult to extrapolate from these results to clinical situations where antifungal agents are used against *Alternaria* spp.. However, our results suggest that drug-induced changes in the cell membrane could affect the outcome of fungal cell interactions with immune cells *in vivo*. Our model circumvents a major limitation for studying the hyphal phase of filamentous fungi, namely their destructiveness for macrophages. Therefore, we strongly believe that the surrogate described here might become a powerful approach to study the interaction of filamentous fungi with host cells, in particular with the respiratory airways, the main portal of entry of fungal spores, but also of hyphal fungal particles when the environmental mycelia are disaggregated, leading to hypersensitization and allergic reactions. This deserves further future work unraveling the PRR involved, using knockout mutant cell lines for the main receptors recognizing the fungal cell wall components and other cell lines such as human macrophages and respiratory epithelial cells, but also longer periods of interaction, mimicking prolonged chronic exposure leading to fungal sensitization.

In summary, the main conclusion redrawn from this work, using CWNPs as a surrogate for the interaction of hyphal fungal cell walls with host cells (Fig. 7), is that DHN-melanin synthesis inhibition prompts macrophages to readily undergo cell death with signs of autophagy but do not trigger immediate production of immune determinants such as TNF-α; caspofungin by modifying β(1,3)-glucan synthesis and exposure in the hyphal cell wall, led to prompt release of TNF-α and IL-1β but not to immediate cell death; the changes introduced by caspofungin also led to the higher hydrolysis of chitin inside intracellular macrophage compartments and secretion to the extracellular milieu; CWNPs with more chitin, more melanin, and less glucan had a delayed macrophage response and a delayed internalization.

## MATERIALS AND METHODS

### Cells and culture conditions

*A. infectoria* strain was obtained from CBS-KNAW Fungal Biodiversity Center, Utrecht, The Netherlands (CBS 137.9). Fungi were stored at −80°C. *A. infectoria* was grown for at least during 2 weeks in Potato Dextrose Agar (Difco, BD, New Jersey; USA) at 30°C with alternating 16 h light and 8 h dark (day-night) cycles under a blacklight lamp (TL-D 18W BLB; Philips). To obtain a conidia suspension, *A. infectoria* mycelial mats were harvested by scrapping with liquid Yeast Malt Extract [YME: 4% yeast extract (Panreac-Cultimed, Alicante, Spain), 10% malt extract (Bioscience, San Diego, CA, USA), and 10% glucose (Sigma-Aldrich, St. Louis, MO, USA)]. Then, several Erlenmeyer flasks containing YME were inoculated with $1 \times 10^6$ conidia and cultured at 30°C with constant orbital shaking at 120 rpm (with the day-night light cycle described above). Liquid cultures were supplemented with pyroquilon [1,2,5,6-tetrahydropyrrolo (3,2,1,-ij)quinolin-4-one; 50 µg/mL; Sigma-Aldrich, St. Louis, MO, USA], nikkomycin Z (0.5 µg/mL; Sigma-Aldrich, St. Louis, MO, USA), or caspofungin (1 µg/mL), a gift from Merck & Co, Inc., Rahway, NJ (material transfer agreement no. 37006). The drug concentration used was based on our previous studies in which, using a standard EUCAST procedure, it was determined the minimal effective concentrations for nikkomycin (21) and for caspofungin (18). Using an agar assay (21), it was determined that pyroquilon at 25 µg/mL led to a mycelia color change that indicated that melanin biosynthesis was inhibited. The fungal mycelium was harvested through filtration with a steel filter, washed four times with distilled water, and frozen at −80°C.

The mouse macrophage cell line RAW 264.7 was obtained from the European Collection of Cell Cultures (ECACC catalog number 91062702; Salisbury, UK) and maintained in Dulbecco Minimal Essential Medium (DMEM; Sigma-Aldrich, St. Louis, MO, USA) supplemented with 10% non-inactivated fetal bovine serum (Life Technologies, Paisley, UK), 10 mM HEPES, 12 mM sodium bicarbonate, and 11 mg/mL sodium pyruvate

(Sigma-Aldrich, St. Louis, MO, USA) at 37°C in a humidified atmosphere with 5% $CO_2$. The medium was changed every 2 days, until ~70% of confluence was reached. All the experiments were performed in cells under the 15th generation to avoid unresponsiveness of the cells (40).

## Quantification of fungal cell wall chitin and β (1,3)-glucan

The chitin content of *A. infectoria* cell wall grown in the presence of nikkomycin Z, pyroquilon, or caspofungin cell wall was quantified based on measurement of glucosamine (CalBiochem, Merck, Darmstadt, Germany), which is released by acid hydrolysis of purified cell walls. The quantification of the β (1,3)-glucan was accomplished using the Aniline Blue Assay (diammonium salt, Sigma-Aldrich, St. Louis, MO, USA) with minor changes, as described previously (23).

## Preparation of *A. infectoria* CWNPs

Lyophilized mycelium was hydrated with a few drops of distilled water and ground in liquid nitrogen. The mycelium powder was washed with 10 mM Tris-HCl, pH 8.0, 1 mM EDTA (TE), and the pellet was resuspended with TE plus a protease inhibitor cocktail (Sigma-Aldrich, St. Louis, MO, USA). Acid-washed beads (425–600 µm) (Sigma-Aldrich St. Louis, MO, USA) were added and then used for cell disruption with Magna Lyser (Roche, Mannheim, Germany) by applying four cycles of 4,800 rpm for 20 s (with 30 s intervals on ice between each cycle of cell disruption). Supernatants were collected; TE plus a protease inhibitor cocktail was added for a second cycle of cell disruption. The supernatants were collected, and the remaining pellet was washed six more times with the support of a vortex between each wash step. All supernatants were combined. The sum of supernatants obtained during this process was centrifuged at 4,800 *g* for 15 min, and the pellet was resuspended in TE, followed by centrifugation at 3,000 *g* for 5 min. The final supernatant was collected and dried overnight at 100°C on glass test tubes. Finally, CWNPs were resuspended in distilled water and conserved at −20°C. Before use, the CWNPs were centrifuged at 4,800 *g* for 15 min, the pellet resuspended in RPMI medium (Sigma-Aldrich, St. Louis, MO, USA), and sonicated by ultrasound.

## Characterization of the CWNPs

To characterize the CWNPs, particle and zeta potential were measured by DLS and electrophoretic light scattering, respectively, using the DelsaTM Nano C Particle Analyser (Beckman Coulter) with software version 2.31/2.03. To calibrate the equipment, a standard control was used (Otsuka Electronics, Osaka, Japan). The size was confirmed by TEM of particles suspended in water performed using a JEOL JEM 1400, 120 kV (JEOL, Peabody, MA, USA), placing a drop of the sample in a mesh grid which was dried out before visualization.

Flow cytometry analysis was performed to quantify the number of particles and to study the distribution of the populations of CWNPs. It was performed on Partec CyFlow space instrument. The CWNPs were prepared as described previously and labeled with 0.5 µg/mL fluorescein (FITC; Alexa Fluor 488 dye, ThermoFisher Scientific) during a 2 h incubation. Before taking the samples to the cytometer, these were again sonicated. The software used was FloMax 2.62.

## Interaction of CWNPs with macrophages

For macrophage interaction with CWNPs, RAW 264.7 was plated in 24- or 96-well plates (Corning, NY, USA) at a cell density of $1.25 \times 10^5$ cells/mL of RPMI and kept at 37°C in a 5% $CO_2$ atmosphere until ~70% of confluence. The expected macrophage density at the day of interaction was approximately $2.5 \times 10^5$ cells/mL.

The interaction of CWNPs with macrophages was characterized by TEM. After 3 h interaction, the culture medium was removed from the wells, and the adherent cells were washed with cold phosphate-buffered saline (PBS). These cells were fixed with 2.5% glutaraldehyde in 0.1 M sodium cacodylate buffer (pH 7.2) for 2 h. Post fixation was performed using 1% osmium tetroxide for 1 h. Cells were removed by scratching from the support using the pipette tip and washed twice in buffer, followed by two washing steps with buffer and three washing steps in distilled water. Aqueous uranyl acetate (1%) was added to the cells for 1 h in the dark for contrast enhancement. After washing in distilled water, samples were dehydrated in a graded ethanol series (30%–100%), impregnated, and embedded in Epoxy resin (Fluka Analytical, Buchs, Switzerland). Ultrathin sections (~70 nm) were mounted on copper grids and stained with lead citrate 0.2% for 7 min. Observations were carried out on an FEI-Tecnai G2 Spirit Bio Twin at 100 kV.

Fluorescence microscopy was used to study morphological changes on macrophages during the interaction with the different nanoparticles and the cellular localization of the CWNPs. Before the interaction, CWNPs were labeled with 0.1% CFW (Fluorescent Brightener 28, Sigma-Aldrich, USA) during 30 min in the dark, at room temperature. Macrophages were labeled with LysoTracker Red (Red DND-99, Invitrogen, Molecular Probes reference) during 30 min, at 37°C in a humidified atmosphere, with 5% $CO_2$. For the interaction, the concentration of CWNPs was estimated by flow cytometry, and the particles were added to the cells in an equal amount to a number of macrophages.

After 0.5, 1.5, 3, and 6 h of co-incubation, the plates were put on ice, and the cover slips were washed twice with cold PBS. The cells were fixed with 4% paraformaldehyde (Sigma-Aldrich, St. Louis, MO, USA) in PBS for 15 min at room temperature and then washed three times with PBS. The coverslips were mounted on glass sides, using fluorescent mounting medium DAKO (Luso Palex Medical, Barcelona, Spain). Cell imaging was performed on a Carl Zeiss LSM 710 Confocal Microscope, using a 63× Plan-ApoChromat (NA 1.4) oil objective. Image analysis was achieved using Fiji software (41).

## Macrophage viability test

The viability of RAW 264.7 macrophages was quantified upon incubation with *A. infectoria* CWNPs. To perform the viability test, the macrophages were collected by scrapping and stained with 4% trypan blue (T8154, Sigma) diluted in PBS. The number of cells not stained by trypan blue corresponding to cells with intact cytoplasmic membranes was counted using a Neubauer chamber.

## TNF-α gene expression and release of TNF-α and IL-1β

To quantify the relative expression of TNF-α gene in RAW 264.7 cells, a real-time PCR approach was performed, using the GAPDH rRNA gene as a reference gene. The interaction assays were carried out as described previously, although RAW 264.7 cells were plated in six-well plates. After 3 h of co-incubation, the six-well plates were put on ice, and the cells were scraped and transferred to ice-cold RNase-free Eppendorf tubes. After centrifugation (10,000 rpm for 5 min at 4°C), RNA extraction was performed with the NucleoSpin RNA Kit protocol (Macherey-Nagel, Düren, Germany) according to the manufacturer's instructions. RNA concentration was determined from the A260/280 value on a NanoDrop 2000 (Thermo Fisher Scientific, Rockford, IL, USA). Reverse transcription of total RNA was processed into cDNA using the Transcriptor First Strand cDNA synthesis Kit (Roche, Mannheim, Germany) according to the kit instructions. To quantify the relative gene expression of TNF-α, we proceeded to real-time quantitative RT-PCR, using the SsoFastEva Green Supermix (BioRad Laboratories, Inc, Hercules, CA, USA) according to the manufacturer's instructions. Primers used were: TNF-α, forward (5-CATGATCCGCGACGTGGAACTG-3′) and reverse (5′-AGAGGGAGGCCATTTGGGAACT-3′) (40) and, GAPDH forward (5′-GTCTTCACCACCATGGAGA-3′) and reverse (5′-CCAAAGTTGT CATGGATGACC-3′) (65). Gene amplification for quantification of relative expression was

analyzed based on the ratio of Ct values with the normalization against GAPDH, using the $2^{-\Delta\Delta Ct}$ method (66).

To quantify the TNF-α and IL-1β cytokine release by RAW 264.7, cells the interaction assays were carried out as described previously. After 6 h of co-incubation, the plates were put on ice, and the supernatants were collected. The supernatants were centrifuged at 12,000 rpm at 4°C for 10 min, resuspended in PBS, and were stored at − 80°C for cytokine quantification using the commercial ELISA kit (BioLegend) according to the manufacturer's instructions.

## Statistical analysis

Data were analyzed using one-way analysis of variance, followed by Dunnett's *t* test *post hoc* analysis to compare treated samples with untreated control using Prism (version 8) software (GraphPad Software, Inc., La Jolla, CA, USA). Data are presented as the means ± SEMs or SDs, and differences were considered significant at *P* values of <0.05. At least three samples were used for three independent experiments.

## ACKNOWLEDGMENTS

This research was funded by FEDER under Programme–COMPETE 2020 and by Foundation for Science and Technology (FCT; UIDB/04539/2020, UIDP/04539/2020 and LA/P/0058/2020). This work was also financed by the European Regional Development Fund (ERDF), through the Centro 2020 Regional Operational Programme, and the Portuguese national funds via FCT, under project ViraVector (CENTRO-01-0145-FEDER-022095), and the program Pepita (FMUC/Santander). Chantal Fernandes thankful financial support by 10.54499/DL57/2016/CP1448/CT0025.

Conceptualization: T.G., D.A., M.C.-A., and C.F. Methodology: D.A., C.F., R.D., M.C.-A., A.C., O.B., and L.R. Software: D.A. and L.R. Writing—original draft preparation: D.A., C.F., A.C., and T.G. Writing—review and editing: T.G. and A.C. Supervision: T.G. Funding acquisition: T.G. All authors have read and agreed to the published version of the manuscript.

## AUTHOR AFFILIATIONS

[1]Univ Coimbra, CNC-UC—Center for Neuroscience and Cell Biology of the University of Coimbra, Coimbra, Portugal

[2]CIBB—Center for Innovative Biomedicine and Biotechnology, University of Coimbra, Coimbra, Portugal

[3]Univ Coimbra, FFUC—Faculty of Pharmacy, University of Coimbra, Coimbra, Portugal

[4]Life and Health Sciences Research Institute (ICVS), School of Medicine, University of Minho, Braga, Portugal

[5]ICVS/3B's—PT Government Associate Laboratory, Braga/Guimarães, Portugal

[6]Department of Molecular Microbiology and Immunology, Johns Hopkins Bloomberg School of Public Health, Baltimore, Maryland, USA

[7]Univ Coimbra, FMUC—Faculty of Medicine, University of Coimbra, Coimbra, Portugal

## AUTHOR ORCIDs

Agostinho Carvalho http://orcid.org/0000-0001-8935-8030
Arturo Casadevall http://orcid.org/0000-0002-9402-9167
Chantal Fernandes http://orcid.org/0000-0001-7865-9946
Teresa Gonçalves http://orcid.org/0000-0001-9347-0535

## FUNDING

| Funder | Grant(s) | Author(s) |
|---|---|---|
| FEDER-Compete-Fundação para a Ciência e Tecnologia | POCI-01-0145-FEDER-007440 | Chantal Fernandes |
| | | Teresa Gonçalves |
| Centro2020 | ViraVector (CENTRO-01-0145-FEDER-022095) | Chantal Fernandes |
| | | Teresa Gonçalves |

## AUTHOR CONTRIBUTIONS

Daniela Antunes, Conceptualization, Methodology, Software, Writing – original draft | Rita Domingues, Methodology | Mariana Cruz-Almeida, Conceptualization, Methodology | Lisa Rodrigues, Methodology, Software | Olga Borges, Methodology | Agostinho Carvalho, Methodology | Arturo Casadevall, Writing – original draft, Writing – review and editing | Chantal Fernandes, Conceptualization, Methodology, Writing – original draft | Teresa Gonçalves, Conceptualization, Funding acquisition, Supervision, Writing – original draft, Writing – review and editing

## ADDITIONAL FILES

The following material is available online.

Open Peer Review

**PEER REVIEW HISTORY (review-history.pdf).** An accounting of the reviewer comments and feedback.

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
