## [Reviewer comments · Microbiology Spectrum]

Microbiology Spectrum

Cell wall nanoparticles from hyphae of *Alternaria infectoria* grown with caspofungin, nikkomycin or pyroquilon trigger different activation profiles in macrophages

Daniela Antunes, Rita Domingues, Mariana Cruz-Almeida, Lisa Rogrigues, Olga Borges, Agostinho Carvalho, Arturo Casadevall, Chantal Fernandes, and Teresa Goncalves

Corresponding Author(s): Teresa Goncalves, Universidade de Coimbra Faculdade de Medicina

Review Timeline:

Submission Date:	March 11, 2024
Editorial Decision:	April 28, 2024
Revision Received:	July 14, 2024
Accepted:	August 2, 2024

Editor: Renato Kovacs

Reviewer(s): Disclosure of reviewer identity is with reference to reviewer comments included in decision letter(s). The following individuals involved in review of your submission have agreed to reveal their identity: Sumanun SUWUNNAKORN (Reviewer #3)

Transaction Report:

DOI: <https://doi.org/10.1128/spectrum.00645-24>

Re: Spectrum00645-24 (Cell wall nanoparticles from hyphae of *Alternaria infectoria* grown with caspofungin, nikkomycin or pyroquilon trigger different activation profiles in macrophages)

Dear Prof. Teresa Gonçalves:

Thank you for the privilege of reviewing your work. Below you will find my comments, instructions from the Spectrum editorial office, and the reviewer comments.

Revision Guidelines

Sincerely,
Renato Kovacs
Editor
Microbiology Spectrum

Reviewer #1 (Comments for the Author):

The manuscript from Antunes et al, utilizes a clever system for understanding the interactions of cell wall components of *Alternaria infectoria* with macrophages. The authors generated cell wall nanoparticles from organisms that had been treated with drugs that blocked production of beta glucan, chitin, or melanin. They characterized the cell walls of the organisms treated with the drugs, then made the nanoparticles, and used them in culture with a macrophage cell line. Macrophage viability and morphology, engulfment of particles, and cytokine production were measured using various standard techniques. Results were

interesting in that they demonstrated that modulation of the cell wall has a significant effect on macrophage function. Not addressed though is the pattern recognition receptors that the macrophages may have used to recognize the nanoparticles. For example, in the discussion the authors presume that the caspofungin treated nanoparticles induce cytokine production because beta glucans are more exposed. A knockdown of dectin-1 could solidify that assumption. Overall, the manuscript is well presented and written, though an editor should look at tense which is understandably wrong in some spots. Fig 1D needs axis labels that are readable. Fig 4B shading is hard to discern because the lines are thin.

Reviewer #2 (Comments for the Author):

The manuscript by Antunes et al presents data on the preparation of cell wall nanoparticles (cwnp) from *Alternaria infectoria* mycelium and studied its interactions with RAW264.7 macrophage like cells. The rationale for the use of cwnp is based on the technical challenge presented by the bigger size of fungal mycelium for host-pathogen interaction studies. The strengths of this paper is innovative use of cwnp for fungus-macrophage interaction. Listed below are a there are my specific comments that may help improve this manuscript.

1. The discrepancy in chitin content after treatment of cwnp in figure 1 is justified by compensatory effect of other chitin synthases in *A. infectoria*. How many chitin synthases are present in *A. infectoria*? Perhaps the authors can try to include *A. infectoria* conidia as a control in this assay.
2. The multiplicity of infection of fungus-macrophages is not presented in the methods.
3. it is very difficult to see the CFW stained cwnp (blue halo) in figure 3.
4. The killing of macrophages after 6h by cwnp is in sharp contrast of the previous work by the same group with *A. infectoria* conidia. A better description and discussion of these finding will be very helpful in understanding the mechanism by which macrophages are getting killed by cwnps.

Minor comments:

line 46, please change to desegregated

line 52, please change overcome to overcame

Reviewer #3 (Comments for the Author):

The authors have used cell nanoparticles (CWNP) from *Alternaria infectoria* hyphae grown in the presence of cell wall perturbing agents caspofungin, nikkomycin, and pyroquilon to interact with the RAW264.7 mouse macrophage cell line. CWNPs with altered cell wall compositions leads to different responses by macrophages using various methods.

The text of the manuscript is easy to follow. But I have the comments that should be addressed by the authors to improve the quality of the manuscript as below:

1. The number of infections due to resistant strains of *A. infectoria* and other resistant species has not been discussed or tested in manuscript.
2. Did the authors determine concentrations that revealed in vitro growth difference in addition to a significant cell wall component change? Did you have the data for spot assay on agar plates containing drugs.
3. There were different concentrations (0.5 µg/ml or 1 µg/ml) of Nikkomycin used in this study. Is there any reason for that? or which one is correct?
4. Have the authors try to test these CWNPs to interact with THP-1 or U937 human macrophage cell line?
5. In Figure 1, the authors could indicate in Y axis "Glucan and chitin content" instead of "Glucan and glucosamine", and explain in figure legend that "chitin content is expressed as a percentage of glucosamine released per dry weight of tissue.
6. Trypan blue is not ideal method for determining macrophage viability. However, the authors used this method to check viable cells. The authors should use flow cytometry or more precise methods for viability of macrophages.
7. The end point at 6 h for the interaction between macrophages and the different CWNPs was too short to see different responses by macrophages.
8. In figure 5, it would be helpful if the authors indicate short description for the arrows within the TEM figure.
9. It is not clear for the receptors on macrophages and different CWNPs for their interaction.

The authors have used cell nanoparticles (CWNP) from *Alternaria infectoria* hyphae grown in the presence of cell wall perturbing agents caspofungin, nikkomycin, and pyroquilon to interact with the RAW264.7 mouse macrophage cell line. CWNP with altered cell wall compositions leads to different responses by macrophages using various methods.

The text of the manuscript is easy to follow. But I have the comments that should be addressed by the authors to improve the quality of the manuscript as below:

1. The number of infections due to resistant strains of *A. infectoria* and other resistant species has not been discussed or tested in manuscript.
2. Did the authors determine concentrations that revealed *in vitro* growth difference in addition to a significant cell wall component change? Did you have the data for spot assay on agar plates containing drugs.
3. There were different concentrations (0.5 µg/ml or 1 µg/ml) of Nikkomycin used in this study. Is there any reason for that? or which one is correct?
4. Have the authors try to test these CWNP to interact with THP-1 or U937 human macrophage cell line?
5. In Figure 1, the authors could indicate in Y axis "Glucan and chitin content" instead of "Glucan and glucosamine", and explain in figure legend that "chitin content is expressed as a percentage of glucosamine released per dry weight of tissue.
6. Trypan blue is not ideal method for determining macrophage viability. However, the authors used this method to check viable cells. The authors should use flow cytometry or more precise methods for viability of macrophages.
7. The end point at 6 h for the interaction between macrophages and the different CWNP was too short to see different responses by macrophages.

8. In figure 5, it would be helpful if the authors indicate short description for the arrows within the TEM figure.

9. The authors should use the g force rather than RPMs for centrifugation speed.

10. It is not clear for the receptors on macrophages and different CWNPs for their interaction.

RESPONSE TO REVIEWERS

(The line numbers indicated in the Response to reviewers is according to the document "Marked up Manuscript", according to the instructions without figures; the new figures were uploaded and corrected in the "Clean version MS")

Reviewer #1 (Comments for the Author):

The manuscript from Antunes et al, utilizes a clever system for understanding the interactions of cell wall components of Alternaria infectoria with macrophages. The authors generated cell wall nanoparticles from organisms that had been treated with drugs that blocked production of beta glucan, chitin, or melanin. They characterized the cell walls of the organisms treated with the drugs, then made the nanoparticles, and used them in culture with a macrophage cell line. Macrophage viability and morphology, engulfment of particles, and cytokine production were measured using various standard techniques. Results were interesting in that they demonstrated that modulation of the cell wall has a significant effect on macrophage function.

We are in debt to the Reviewer for the time spent reviewing our manuscript, for the forwarded positive evaluation and for the comments that lead us to improve the clarity of the MS.

Not addressed though is the pattern recognition receptors that the macrophages may have used to recognize the nanoparticles. For example, in the discussion the authors presume that the caspofungin treated nanoparticles induce cytokine production because beta glucans are more exposed. A knockdown of dectin-1 could solidify that assumption.

The point raised by the reviewer is pertinent and important in the context of studying the host-pathogen interaction. In this work we wanted to validate an in vitro model for the study of the interaction of hyphal cell wall with host cells, when the cell wall is modified by antifungals. In the Discussion section (**Line 297**) we reinforce this aspect by introducing "hyphal cell wall".

To indicate that we did not verify if beta glucan is more exposed in the casCWNP we added the following sentence (Discussion, Line 457): "Based on previous studies indicating the differential exposure of cell wall components depending on the antifungal used, one can expect that on the casCWNP particles, $\beta(1,3)$ -glucan is presumably more exposed,...".

We can only agree with the Reviewer that the study of the recognition receptors would be an important information. The study, in the current form, opened windows of opportunities to pursue with future work, including the identification of the receptors involved in the recognition of the CWNPs. We thank the reviewer for the suggestion that using a dectin-1 KO would be of great interest to further characterize the interaction of these nanoparticles with host cells and its suitability as a model system. To highlight this important aspect we added the following in the Discussion section (**Line 431**): "This deserves further future work unravelling the Pattern Recognition Receptors (PRR) involved, using knockout mutant cell lines for the main receptors recognizing the fungal cell wall components, other cell lines such as human macrophages and respiratory epithelial cells, but also longer periods of interaction, mimicking prolonged chronic exposure leading to fungal sensitization."

Overall, the manuscript is well presented and written, though an editor should look at tense which is understandably wrong in some spots.

Careful reading and correction of language was done throughout the text.

Fig 1D needs axis labels that are readable.

We believe that the reviewer meant Figure 2D. In this Figure the axis label was modified to increase readability.

Fig 4B shading is hard to discern because the lines are thin.

Figure 4 was modified to increase the readability of the both graph A and B. We also added the x axis label, to Fig 4B.

Reviewer #2 (Comments for the Author):

The manuscript by Antunes et al presents data on the preparation of cell wall nanoparticles (cwnp) from Alternaria infectoria mycelium and studied its interactions with RAW264.7 macrophage like cells. The rationale for the use of cwnp is based on the technical challenge presented by the bigger size of fungal mycelium for host-pathogen interaction studies. The strengths of this paper is innovative use of cwnp for fungus-macrophage interaction. Listed below are a there are my specific comments that may help improve this manuscript.

We very much thank the Reviewer for the time spent in carefully revising our MS and for the numerous suggestions provided, which considerably increased the rigor and correctness of the information provided.

1. The discrepancy in chitin content after treatment of cwnp in figure 1 is justified by compensatory effect of other chitin synthases in A. infectoria. How many chitin synthases are present in A. infectoria? Perhaps the authors can try to include A. infectoria conidia as a control in this assay.

Previously, we described that *A. infectoria* holds 8 different chitin synthases genes, whose expression is differently modulated by either caspofungin or nikkomycin (Ref 23 - Fernandes et al., 2014; 10.1128/AAC.02647-13).

Before we described in detail how *A. infectoria* conidia interact with RAW 264.7 macrophage cells [mainly in Ref 23 - Fernandes et al., 2014 (10.1128/AAC.02647-13); also in Ref 40 -Almeida et al., 2019 (10.1007/s11046-019-00339-6)]; in the current work we restricted our study to hyphal cell wall nanoparticles as we describe in the manuscript at the 3rd paragraph of the Discussion section. Including again the interaction of RAW264.7 with conidia would be overlapping with the previous studies.

2. The multiplicity of infection of fungus-macrophages is not presented in the methods.

In the section Material and Methods, Line **548**, we describe that “For the interaction, the CWNPs concentration was estimated by flow cytometry and were added to the cells in an equal amount to number of macrophages.”. We did not use the term “MOI – multiplicity of infection” because we were exposing host cells to nanoparticles not to infectious agents. However, if the reviewer thinks the inclusion of MOI is appropriate, we will introduce that in the text.

3. it is very difficult to see the CFW stained cwnp (blue halo) in figure 3.

Figure 3 is a panel that includes a high number of photos to illustrate the interaction of macrophages with the CWNPs obtained under different conditions and using a differential fluorescence microscopy method, at 30 min, 1.5h, 3h and 6h. We agree with the reviewer that it is a bit difficult to see some details. For that reason, we included, as Supplementary Materials, representative 3D projections (videos) of macrophages interacting with CWNPs *A. infectoria* (ctCWNPs, casCWNPs, nikkoCWNPs, PyrCWNPs). Below we post a frame of the video illustrating the blue halo in the case of casCWNPs:

4. *The killing of macrophages after 6h by cwnp is in sharp contrast of the previous work by the same group with A. infectoria conidia. A better description and discussion of these finding will be very helpful in understanding the mechanism by which macrophages are getting killed by cwnps.*

We acknowledge this comment by the reviewer. Based on this we modified the Discussion to highlight the difference between conidia (previously published) and the CWNPs interaction with macrophages.

We changed the paragraph beginning at Line **357** to:

“The early (first 6 hours) interaction of *A. infectoria* conidia with RAW264.7 macrophages did not change macrophage’s viability, as previously indicated by us (REF 40 - Almeida et al., 2019; 10.1007/s11046-019-00339-6). Now, it is reported that despite the ctCWNP not influencing macrophage viability initially, after 6 h interaction, CWNPs from fungi grown in the presence of pyroquilon, nikkomycin Z and caspofungin all reduced macrophage viability. So, the structure/components of the hyphal ctCWNP, when interacting with macrophages, most probably behave similarly to conidia, with a delayed mild response from the host cells. Otherwise, when the fungal hyphal cell wall structure is affected by antifungals directed to the synthesis of *A. infectoria* cell wall components such as chitin, beta glucan and DHN-melanin, PAMPs in the resulting CWNP exert a more exacerbated response by macrophages which leads to loss of cell viability.”

Minor comments:

line 46, please change to desegregated

Changed accordingly

line 52, please change overcome to overcame

Changed accordingly

Reviewer #3 (Comments for the Author):

The authors have used cell nanoparticles (CWNP) from Alternaria infectoria hyphae grown in the presence of cell wall perturbing agents caspofungin, nikkomycin, and pyroquilon to interact with the RAW264.7 mouse macrophage cell line. CWNP with altered cell wall compositions leads to different responses by macrophages using various methods.

We are in debt to the Reviewer for the time spent reviewing our manuscript, for the forwarded positive evaluation and for the questions and comments that lead us to improve the MS.

The text of the manuscript is easy to follow. But I have the comments that should be addressed by the authors to improve the quality of the manuscript as below:

1. The number of infections due to resistant strains of A. infectoria and other resistant species has not been discussed or tested in manuscript.

Several cases of *Alternaria* spp. infections have been published, although the in vitro antifungal susceptibility data were only reported in a few of these. Pujol and collaborators (Pujol et al., 2000; <https://doi.org/10.1093/jac/46.2.337>) described that *Alternaria* spp. are resistant to flucytosine and the activity of amphotericin B and fluconazole is variable. However, in vivo treatments with these drugs showed positive outcome. Our research group, together with clinicians, reported, in 2008 (Hipólito et al., 2008; 10.1007/s10096-008-0623-2), a resilient central nervous system infection with *A. infectoria*. Recently, we revised all this information (Ref 1: Fernandes et al., 2023; DOI: 10.1093/femsre/fuad061)

In the manuscript we added the following:

Line 69 –) ... and the information regarding resistance to antifungals used in human health to treat *Alternaria* spp. is scarce (revised in 1).”

2. Did the authors determine concentrations that revealed in vitro growth difference in addition to a significant cell wall component change? Did you have the data for spot assay on agar plates containing drugs.

This important aspect now raised by the reviewer was demonstrated by us in previous publications. Instead of a spot assay, we used a distinct parameter for caspofungin and nikkomycin Z. We determined the minimum effective concentration (MEC), as recommended for echinocandins in the updated EUCAST standards (EUCAST E.DEF 9.4 March 2022). The MEC defines the lowest concentration of the drug yielding morphological alterations. In *A. infectoria*, caspofungin and nikkomycin Z lead to the formation of abnormal balloon-like cells. Using the standard procedure we measured the nikkomycin (Ref 21 - Fernandes et al., 2015; 10.1128/AAC.02190-15) and caspofungin (Ref 18 - Anjos et al., 2012; 10.3109/13693786.2012.675525) MECs, concentrations at which the morphological alterations occurred. Different concentration of pyroquilon were tested on agar (REF 21 - Fernandes et al., 2015; 10.1128/AAC.02190-15), observing that at 25 ug/ml of pyroquilon, the mycelial mat changed from black to a reddish brown color, and a typically reddish pigment surrounding the colony indicated that melanin biosynthesis was inhibited. Moreover, previously, we also described:

- chitin synthase genes identification and expression, alteration of the hyphal and colonial morphology, modulation of the chitin and glucan cell wall components in response to caspofungin and nikkomycin (Ref 23 - Fernandes et al., 2014; 10.1128/AAC.02647-13).
- modulation of melanin accumulation in response to caspofungin and nikkomycin (Ref 21 - Fernandes et al., 2015; 10.1128/AAC.02190-15).

Throughout the manuscript the previous results that support this work were mentioned. Some examples:

Line 387 – “Previous studies from our research group have focused on the cell wall structure and its modulation by antifungals of common environmental fungi, *Alternaria* sp., with particular interest in *A. infectoria*, which is an opportunistic agent of human fungal infection and of fungal sensitization and asthma (18,21,23).”

Line 417 – “*A. infectoria* grown in the presence of caspofungin leads to increased melanin content and decrease in β (1,3)-glucan while the chitin cell wall levels remain unchanged (21,23), and the mycelia become darker. Paradoxically, when grown in the presence of nikkomycin Z, a chitin synthase inhibitor, the amount of chitin in the cell wall increased.”

Line 469 – “For *A. infectoria*, nikkomycin Z leads to higher cell wall chitin content and, concomitantly, to higher DHN-melanin cell wall content (21,59).”

Line 477 – “The pyrCWNP, prepared from mycelia grown with pyroquilon, an inhibitor of DHN-melanin synthesis and an inducer of lower cell wall chitin content (21).”

In order to clarify better these important aspects in the manuscript we added:

Line 465 (Materials and Methods section): “The drugs concentration used was based on our previous studies in which, using a standard EUCAST procedure, it was determined the Minimal Effective Concentrations (MECs) for nikkomycin (21) and for caspofungin (18).

Using an agar assay (21) it was determined that pyroquilon at 25 ug/ml led to a mycelia color change that indicated that melanin biosynthesis was inhibited.”

3. There were different concentrations (0.5 µg/ml or 1 µg/ml) of Nikkomycin used in this study. Is there any reason for that? or which one is correct?

The correct concentration of Nikkomycin is 0.5 µg/ml. Throughout the MS this error was corrected.

4. Have the authors try to test these CWNPs to interact with THP-1 or U937 human macrophage cell line?

No, we never tried these other cell lines, because our model host cells are RAW264.7 macrophage cell line. As future work, it is our intention to extend these assays to human macrophages, neutrophils and respiratory epithelial cells. Some preliminary assays give promising results but we are expecting financing to proceed.

5. In Figure 1, the authors could indicate in Y axis "Glucan and chitin content" instead of "Glucan and glucosamine", and explain in figure legend that "chitin content is expressed as a percentage of glucosamine released per dry weight of tissue."

Modified accordingly.

6. Trypan blue is not ideal method for determining macrophage viability. However, the authors used this method to check viable cells. The authors should use flow cytometry or more precise methods for viability of macrophages.

Trypan blue assays are commonly used in infection experiments, in combination with other tests or by itself, as a good indicator of membrane integrity, since they are simple and inexpensive. The consistent trend of the experiments performed, with statistical differences and acceptable error values, lead us to consider that the results obtained with the trypan blue assay were sufficiently accurate for the objective of our study. Nevertheless, we agree with the Reviewer's observation, since, in fact, there are other viability tests that can be used (MTT, XTT, alamarBlue, LDH, ATP luminescent assays...), some of which are considered more reliable and sensitive to obtain quantitative results, but that also have limitations. Undoubtedly, the association of flow cytometry to these viability tests increases the accuracy of the absolute cell viability. Consistent with this, we added a comment in the Discussion section (Line 354) stating that: "In this work we quantified macrophages viability using a trypan blue exclusion assay and, although there are other quantification methodologies (51), and the association of flow cytometry (52) increases the accuracy of using trypan blue assay, the robustness and consistency of the results justifies its validation.

Added references:

51. Cai Y, Prochazkova M, Kim YS, Jiang C, Ma J, Moses L, Martin K, Pham V, Zhang N, Highfill SL, Somerville RP, Stroncek DF, Jin P. 2024. Assessment and comparison of viability assays for cellular products. *Cytotherapy*. 26:201-9.

52. Strober W. 2015. Trypan Blue Exclusion Test of Cell Viability. *Curr Protoc Immunol*. 111:A3.B.1-A3.B.3.

7. The end point at 6 h for the interaction between macrophages and the different CWNPs was too short to see different responses by macrophages.

The end point of 6 h was selected based on our previous work studying the early interaction of conidia with RAW264.7 macrophages [Ref 23 - Fernandes et al., 2014 (10.1128/AAC.02647-13); Ref 40 -Almeida et al., 2019 (10.1007/s11046-019-00339-6)]. As pointed out in the Response to Reviewer #1, in this work we wanted to validate an

in vitro model for the study of the interaction of hyphal cell wall with host cells, when the cell wall is modified by antifungals. In the Discussion section (Line 371) we reinforce this aspect by introducing “hyphal cell wall”.

The study, in the current form, opened windows of opportunities to pursue with future work, including long term exposure to fungal cell walls (extremely relevant when studying sensitization to fungi, for example). We added the following sentence to the Discussion section:

Line **431** – “This deserves further future work unravelling the PRR involved and using knockout mutant cell lines for the main receptors recognizing the cell wall components, other cell lines such as human macrophages and respiratory epithelial cells, but also longer periods of interaction, mimicking prolonged chronic exposure leading to fungal sensitization.”

8. In figure 5, it would be helpful if the authors indicate short description for the arrows within the TEM figure.

Changed accordingly. We also changed all the arrows to black since we introduced a short description. Some of the white arrows were difficult to observe. The legend and text were changed accordingly.

9. It is not clear for the receptors on macrophages and different CWNPs for their interaction.

The point raised by the reviewer (and by reviewer #1) is pertinent and important in the context of studying the host-pathogen interaction. In this work we wanted to validate an in vitro model for the study of the interaction of hyphal cell wall with host cells, when the cell wall is modified by antifungals. In the Discussion section (Line 371) we reinforce this aspect by introducing “hyphal cell wall”.

We can only agree with the Reviewer that the study of the involvement of the recognition receptors would be important information. The study, in its current form, opened windows of opportunities to pursue with future work, including the identification of the receptors involved in the recognition of the CWNPs. This and other further future work would be dependent on financing.

To highlight this important aspect we added the following in the Discussion section (Line **431**): “. This deserves further future work unravelling the Pattern Recognition Receptors (PRR) involved, using knockout mutant cell lines for the main receptors recognizing the fungal cell wall components and other cell lines such as human macrophages and respiratory epithelial cells, but also longer periods of interaction, mimicking prolonged chronic exposure leading to fungal sensitization.”

Re: Spectrum00645-24R1 (Cell wall nanoparticles from hyphae of *Alternaria infectoria* grown with caspofungin, nikkomycin or pyroquilon trigger different activation profiles in macrophages)

Dear Prof. Teresa Goncalves:

Your manuscript has been accepted, and I am forwarding it to the ASM production staff for publication. Your paper will first be checked to make sure all elements meet the technical requirements. ASM staff will contact you if anything needs to be revised before copyediting and production can begin. Otherwise, you will be notified when your proofs are ready to be viewed.

Sincerely,
Renato Kovacs
Editor
Microbiology Spectrum

RESPONSE TO REVIEWERS

(The line numbers indicated in the Response to reviewers is according to the document)

Reviewer #3 (Comments for the Author):

The authors have addressed all my points as below in the revised version of manuscript. Therefore, I have no further comments.

The authors have used cell nanoparticles (CWNP) from Alternaria infectoria hyphae grown in the presence of cell wall perturbing agents caspofungin, nikkomycin, and pyroquilon to interact with the RAW264.7 mouse macrophage cell line. CWNPs with altered cell wall compositions leads to different responses by macrophages using various methods.

We are in debt to the Reviewer for the time spent reviewing our manuscript, for the forwarded positive evaluation and for the questions and comments that lead us to improve the MS.

The text of the manuscript is easy to follow. But I have the comments that should be addressed by the authors to improve the quality of the manuscript as below:

1. The number of infections due to resistant strains of A. infectoria and other resistant species has not been discussed or tested in manuscript.

Several cases of *Alternaria* spp. infections have been published, although the in vitro antifungal susceptibility data were only reported in a few of these. Pujol and collaborators (Pujol et al., 2000; <https://doi.org/10.1093/jac/46.2.337>) described that *Alternaria* spp. are resistant to flucytosine and the activity of amphotericin B and fluconazole is variable. However, in vivo treatments with these drugs showed positive outcome. Our research group, together with clinicians, reported, in 2008 (Hipólito et al., 2008; 10.1007/s10096-008-0623-2), a resilient central nervous system infection with *A. infectoria*. Recently, we revised all this information (Ref 1: Fernandes et al., 2023; DOI: 10.1093/femsre/fuad061)

In the manuscript we added the following:

Line **69** –) ... and the information regarding resistance to antifungals used in human health to treat *Alternaria* spp. is scarce (revised in 1).”

2. Did the authors determine concentrations that revealed in vitro growth difference in addition to a significant cell wall component change? Did you have the data for spot assay on agar plates containing drugs.

This important aspect now raised by the reviewer was demonstrated by us in previous publications. Instead of a spot assay, we used a distinct parameter for caspofungin and nikkomycin Z. We determined the minimum effective concentration (MEC), as recommended for echinocandins in the updated EUCAST standards (EUCAST E.DEF 9.4 March 2022). The MEC defines the lowest concentration of the drug yielding morphological alterations. In *A. infectoria*, caspofungin and nikkomycin Z lead to the formation of abnormal balloon-like cells. Using the standard procedure we measured the nikkomycin (Ref 21 - Fernandes et al., 2015; 10.1128/AAC.02190-15) and caspofungin (Ref 18 - Anjos et al., 2012; 10.3109/13693786.2012.675525) MECs, concentrations at which the morphological alterations occurred. Different concentration of pyroquilon were tested on agar (REF 21 - Fernandes et al., 2015; 10.1128/AAC.02190-15), observing that at 25 ug/ml of pyroquilon, the mycelial mat changed from black to a reddish brown color, and a typically reddish pigment surrounding the colony indicated that melanin biosynthesis was inhibited.

Moreover, previously, we also described:

- chitin synthase genes identification and expression, alteration of the hyphal and colonial morphology, modulation of the chitin and glucan cell wall components in response to caspofungin and nikkomycin (Ref 23 - Fernandes et al., 2014; 10.1128/AAC.02647-13).

- modulation of melanin accumulation in response to caspofungin and nikkomycin (Ref 21 - Fernandes et al., 2015; 10.1128/AAC.02190-15).

Throughout the manuscript the previous results that support this work were mentioned. Some examples:

Line 387 – “Previous studies from our research group have focused on the cell wall structure and its modulation by antifungals of common environmental fungi, *Alternaria* sp., with particular interest in *A. infectoria*, which is an opportunistic agent of human fungal infection and of fungal sensitization and asthma (18,21,23).”

Line 417 – “*A. infectoria* grown in the presence of caspofungin leads to increased melanin content and decrease in β (1,3)-glucan while the chitin cell wall levels remain unchanged (21,23), and the mycelia become darker. Paradoxically, when grown in the presence of nikkomycin Z, a chitin synthase inhibitor, the amount of chitin in the cell wall increased.”

Line 469 – “For *A. infectoria*, nikkomycin Z leads to higher cell wall chitin content and, concomitantly, to higher DHN-melanin cell wall content (21,59).”

Line 477 – “The pyrCWNP, prepared from mycelia grown with pyroquilon, an inhibitor of DHN-melanin synthesis and an inducer of lower cell wall chitin content (21).”

In order to clarify better these important aspects in the manuscript we added:

Line **465** (Materials and Methods section): “The drugs concentration used was based on our previous studies in which, using a standard EUCAST procedure, it was determined the Minimal Effective Concentrations (MECs) for nikkomycin (21) and for caspofungin (18). Using an agar assay (21) it was determined that pyroquilon at 25 $\mu\text{g/ml}$ led to a mycelia color change that indicated that melanin biosynthesis was inhibited.”

3. There were different concentrations (0.5 $\mu\text{g/ml}$ or 1 $\mu\text{g/ml}$) of Nikkomycin used in this study. Is there any reason for that? or which one is correct?

The correct concentration of Nikkomycin is 0.5 $\mu\text{g/ml}$. Throughout the MS this error was corrected.

4. Have the authors try to test these CWNP to interact with THP-1 or U937 human macrophage cell line?

No, we never tried these other cell lines, because our model host cells are RAW264.7 macrophage cell line. As future work, it is our intention to extend these assays to human macrophages, neutrophils and respiratory epithelial cells. Some preliminary assays give promising results but we are expecting financing to proceed.

5. In Figure 1, the authors could indicate in Y axis "Glucan and chitin content" instead of "Glucan and glucosamine", and explain in figure legend that "chitin content is expressed as a percentage of glucosamine released per dry weight of tissue."

Modified accordingly.

6. Trypan blue is not ideal method for determining macrophage viability. However, the authors used this method to check viable cells. The authors should use flow cytometry or more precise methods for viability of macrophages.

Trypan blue assays are commonly used in infection experiments, in combination with other tests or by itself, as a good indicator of membrane integrity, since they are simple and inexpensive. The consistent trend of the experiments performed, with statistical differences and acceptable error values, lead us to consider that the results obtained with the trypan blue assay were sufficiently accurate for the objective of our study. Nevertheless, we agree with the Reviewer's observation, since, in fact, there are other viability tests that can be used (MTT, XTT, alamarBlue, LDH, ATP luminescent assays...), some of which are considered more reliable and sensitive to obtain quantitative results, but that also have

limitations. Undoubtedly, the association of flow cytometry to these viability tests increases the accuracy of the absolute cell viability. Consistent with this, we added a comment in the Discussion section (Line **354**) stating that: “In this work we quantified macrophages viability using a trypan blue exclusion assay and, although there are other quantification methodologies (51), and the association of flow cytometry (52) increases the accuracy of using trypan blue assay, the robustness and consistency of the results justifies its validation.

Added references:

51. Cai Y, Prochazkova M, Kim YS, Jiang C, Ma J, Moses L, Martin K, Pham V, Zhang N, Highfill SL, Somerville RP, Stroncek DF, Jin P. 2024. Assessment and comparison of viability assays for cellular products. *Cytotherapy*. 26:201-9.

52. Strober W. 2015. Trypan Blue Exclusion Test of Cell Viability. *Curr Protoc Immunol*. 111:A3.B.1-A3.B.3.

7. The end point at 6 h for the interaction between macrophages and the different CWNPs was too short to see different responses by macrophages.

The end point of 6 h was selected based on our previous work studying the early interaction of conidia with RAW264.7 macrophages [Ref 23 - Fernandes et al., 2014 (10.1128/AAC.02647-13); Ref 40 -Almeida et al., 2019 (10.1007/s11046-019-00339-6)]. As pointed out in the Response to Reviewer #1, in this work we wanted to validate an in vitro model for the study of the interaction of hyphal cell wall with host cells, when the cell wall is modified by antifungals. In the Discussion section (Line 371) we reinforce this aspect by introducing “hyphal cell wall”.

The study, in the current form, opened windows of opportunities to pursue with future work, including long term exposure to fungal cell walls (extremely relevant when studying sensitization to fungi, for example). We added the following sentence to the Discussion section:

Line **431** – “This deserves further future work unravelling the PRR involved and using knockout mutant cell lines for the main receptors recognizing the cell wall components, other cell lines such as human macrophages and respiratory epithelial cells, but also longer periods of interaction, mimicking prolonged chronic exposure leading to fungal sensitization.”

8. In figure 5, it would be helpful if the authors indicate short description for the arrows within the TEM figure.

Changed accordingly. We also changed all the arrows to black since we introduced a short description. Some of the white arrows were difficult to observe. The legend and text were changed accordingly.

9. It is not clear for the receptors on macrophages and different CWNPs for their interaction.

The point raised by the reviewer (and by reviewer #1) is pertinent and important in the context of studying the host-pathogen interaction. In this work we wanted to validate an in vitro model for the study of the interaction of hyphal cell wall with host cells, when the cell wall is modified by antifungals. In the Discussion section (Line 371) we reinforce this aspect by introducing “hyphal cell wall”.

We can only agree with the Reviewer that the study of the involvement of the recognition receptors would be important information. The study, in its current form, opened windows of opportunities to pursue with future work, including the identification of the receptors involved in the recognition of the CWNPs. This and other further future work would be dependent on financing.

To highlight this important aspect we added the following in the Discussion section (Line **431**): “ This deserves further future work unravelling the Pattern Recognition Receptors (PRR) involved, using knockout mutant cell lines for the main receptors recognizing the fungal cell

wall components and other cell lines such as human macrophages and respiratory epithelial cells, but also longer periods of interaction, mimicking prolonged chronic exposure leading to fungal sensitization.”